

# Observations of the summertime atmospheric pollutants vertical distributions and the corresponding ozone production in Shanghai, China

Chengzhi Xing[1#], Cheng Liu[1,2,3,9#*], Shanshan Wang[4*], Ka Lok Chan[5*], Yang Gao[6], Xin Huang[7], Wenjing Su[1], Chengxin Zhang[1], Yunsheng Dong[3], Guangqiang Fan[3], Tianshu Zhang[3], Zhenyi Chen[3], Qihou Hu[3], Hang Su[8,10], Zhouqing Xie[1,2,3,9], Jianguo Liu[2,3]

[1]School of Earth and Space Sciences, University of Science and Technology of China, Hefei, 230026, China
[2]Center for Excellence in Regional Atmospheric Environment, Institute of Urban Environment, Chinese Academy of Sciences, Xiamen, 361021, China
[3]Key Lab of Environmental Optics & Technology, Anhui Institute of Optics and Fine Mechanics, Chinese Academy of Sciences, Hefei, 230031, China
[4]Shanghai Key Laboratory of Atmospheric Particle Pollution and Prevention (LAP[3]), Department of Environmental Science and Engineering, Fudan University, Shanghai, 200433, China
[5]Meteorological Institute, Ludwig-Maximilians Universität München, Munich, Germany
[6]College of Environmental Science and Engineering, Ocean University of China, Qingdao, 266100, China
[7]School of Atmospheric Sciences, Nanjing University, Nanjing, 210093, China
[8]Institute for Environmental and Climate Research, Jinan University, Guangzhou 511443, China
[9]Anhui Province Key Laboratory of Polar Environment and Global Change, USTC, Hefei, 230026, China
[10]Biogeochemistry Department, Max Planck Institute for Chemistry, Mainz, 55020, Germany

[#]This two authors contributed equally

[*]*Correspondence to*: Shanshan Wang (shanshanwang@fudan.edu.cn), Ka Lok Chan (lok.chan@lmu.de), Cheng Liu(chliu81@ustc.edu.cn)

**Abstract.** Ground based Multi-Axis Differential Optical Absorption Spectroscopy (MAX-DOAS) and lidar measurements were performed in Shanghai, China during May 2016 to investigate the summertime atmospheric pollutants vertical distribution. In this study, vertical profiles of aerosol extinction coefficient, nitrogen dioxide ($NO_2$) and formaldehyde (HCHO) concentrations were retrieved from MAX-DOAS measurement using the Heidelberg Profile (HeiPro) algorithm, while vertical distribution of ozone ($O_3$) was obtained from an ozone lidar. Sensitivity study of the MAX-DOAS aerosol profile retrieval shows that the a priori aerosol profile shape has significant influences on the aerosol profile retrieval. Aerosol profiles retrieved from MAX-DOAS measurements with Gaussian a priori demonstrate the best agreements with simultaneous lidar measurements and vehicle-based tethered-balloon observations among all a priori aerosol profiles. MAX-DOAS measured tropospheric $NO_2$ Vertical Column Densities (VCDs) show a good agreement with OMI satellite observations with Pearson correlation coefficient (R) of 0.95. In addition, measurements of the $O_3$ vertical distribution indicate that the ozone productions do not only occur at surface level but also at higher altitudes (about 1.1 km). Planetary boundary layer (PBL) height, horizontal and vertical wind fields information were integrated to discuss the ozone formation



at upper altitudes. The results reveal that enhanced ozone concentrations at ground and upper altitudes are not directly related to horizontal and vertical transportations. Similar patterns of $O_3$ and HCHO vertical distributions were observed during this campaign, which implies that the ozone productions near to the surface and at higher altitudes are mainly influenced by the abundance of volatile organic compounds (VOCs) in the lower troposphere.

## 1 Introduction

Air pollution has become one of the major environmental problems around the world. It is particularly serious in China due the rapid development of economy and industrialization. This problem directly affected the ecological environment and earth's radiation budget. It also has a series of adverse impacts on human health. Regional ozone pollution caused by photochemical reaction during summertime becomes a serious problem in China in the recent years. Previous studies of the $O_3$ vertical distribution and temporal variation of $O_3$ within the boundary layer in northern China indicated the $O_3$ levels are closely correlated with the abundance of $O_3$ precursors (Tang et al., 2017a and 2017b). The major primary $O_3$ precursors in China are nitrogen oxides ($NO_x$), defined as the sum of nitric oxide (NO) and nitrogen dioxide ($NO_2$), and volatile organic compounds (VOCs) (Geng et al., 2007). Nitrogen dioxide and formaldehyde (HCHO) are the important constituents in the atmosphere playing key roles in both tropospheric and stratospheric chemistry (Seinfeld et al., 1998; Chan et al., 2015; Wang et al., 2017). $NO_2$ contributes to the formation of secondary aerosols and participate in the catalytic formation of ozone in the troposphere (Crutzen 1975), while HCHO is one of the most important VOCs which can be used as a proxy for the total VOCs (Sillman, 1995; Duncan et al., 2010).

In the presence of sunlight, VOCs and $NO_x$ contribute together to the formation of ozone in the troposphere (Crutzen, 1975; Seinfeld et al., 1998). The ambient ozone level strongly depends on both the relative and absolute amounts of VOCs and $NO_x$ (Geng et al., 2007; Tang et al., 2009; Tang et al., 2012). Moreover, there are many studies using $HCHO/NO_2$ column ratio as an indicator to determine surface $O_3$-NOx-VOCs sensitivity in previous (Martin et al., 2004; Tang et al., 2012; Mahajan et al., 2015). However, there is a lack of observation for NOx and HCHO vertical distribution in order to investigate the $O_3$ formation and atmospheric chemistry in the lower troposphere. Modelling studies of boundary layer ozone over norther china show different sensitivity of ozone regime in vertical, which is also impacted by the vertical circulation (Tang et al., 2017a and 2017b). In addition, aerosols, particularly fine particles, are one of the major air pollutants in China. They play a key role in the Earth's climate and weather system. The chemical and physical properties of aerosols are strongly dependent on their compositions and sources. Previous studies show that secondary inorganic aerosols (sulfate and nitrate) are the dominating composition of the fine particles in both northern and eastern China (i.e., Shanghai and Beijing) (Du et al., 2011; Zhu et al., 2016). Therefore, we focus on analyzing the secondary formation process of aerosols and photochemical pollution in this study.

Multi-Axis Differential Optical Absorption Spectroscopy (MAX-DOAS) is a passive remote sensing technique measuring spectra of scattered sunlight at different elevation angles. Combining with forward radiative transfer simulations,



MAX-DOAS measurements can provide vertical distribution information of aerosol extinction and trace gases concentration in the lower troposphere (Hönninger and Platt, 2002; Bobrowski et al., 2003; Hönninger et al., 2004; Wagner et al., 2004;

Wittrock et al., 2004; Platt and Stutz, 2008). In the past decade, ground based MAX-DOAS has been widely used for atmospheric aerosol and trace gases measurements (Frieß et al., 2006; Irie et al., 2008; Li et al., 2010; Li et al., 2013). MAX-DOAS measurements are often used to validate satellite observations of atmospheric trace gases, e.g. $NO_2$, $SO_2$ and HCHO (Irie et al., 2008; Kramer et al., 2008; Ma et al., 2013; Wang et al., 2017). Light detection and ranging (lidar) is an active remote sensing measurement technique providing the quantitative range-resolved aerosol parameters. Aerosol vertical

distribution obtained from Lidar measurements are useful for the validation of MAX-DOAS retrieval of aerosol extinction profiles (Irie et al., 2008; Lee et al., 2011). Advanced lidar systems can also provide vertical profiles of different atmospheric species, such as ozone and water vapour.

  Shanghai is one of the four direct-controlled municipalities in China, with population of over 24 million. Owing to the rapid urbanization of Shanghai and its surrounding cities in Yangtze River Delta (YRD) region, air quality has become

deteriorated and much more attractive (Geng et al., 2007; Chan et al., 2015). Shanghai has a large number of vehicles in China. Vehicle emissions were reported to contribute about 35% of the overall $NO_x$ emission in Shanghai (Li et al., 2011; Hao et al., 2011). FengXian, where the experiment was performed, is one of the sub-districts of Shanghai (see Fig. S1) located in the north of Hangzhou gulf. The measurement site is mainly surrounded by agriculture area with only few industrial and traffic emissions. During the measurement campaign, this area was mainly affected by the East Asian

Monsoon with prevailing winds being mainly south-easterly. Previous study shows that ozone pollution is particularly serious in rural areas of Shanghai during summer. Biogenic VOCs was found to be one of the major ozone precursors near the ground surface (Geng et al., 2007). However, there is a lack of observations for the investigation on the ozone formation in vertical.

  In this paper, we present the MAX-DOAS and lidar measurements for the vertical distribution of aerosol and trace

gases in FengXian during May 2016. The vertical profiles of aerosol and $NO_2$ were retrieved and also validated by balloon-based measurements. Vertical distributions of $NO_2$ and HCHO, as well as ozone profile, were used to investigate the ozone formation. In addition, horizontal and vertical fluxes of ozone were calculated from WRF-Chem simulations to estimate the ozone production at different altitudes.

## 2 Measurements and methodology

### 2.1 The MAX-DOAS measurements

  The MAX-DOAS instrument operated at the measurement site in FengXian consists of a telescope, two spectrometers with temperature stabilized at 20°C and a computer acting as controlling and data acquisition unit. The viewing elevation angle of the telescope is controlled by a stepping motor. Scattered sunlight collected by the telescope is redirected by a prism reflector and quartz fibers to the spectrometer for spectral analysis. Two spectrometers (Ocean Optic HR2000+ and a



Maya2000Pro spectrometer) were used to cover both the UV (303-370 nm) and VIS (390-608 nm) wavelength ranges. The full width half maximum (FWHM) spectral resolution of the UV and Vis spectrometers are 0.5 and 0.3 nm, respectively. The field of view (FOV) of the instrument is estimated to be less than 1 °.

During the measurement period, the viewing azimuth direction was adjusted to the north. A full measurement sequence consists of 7 elevation angles, i.e. 3 °, 5 °, 8 °, 10 °, 15 °, 30 ° and 90 °, each with 100 scans. The exposure time is automatically
calculated depending on the intensity of the received scattered sunlight in order to achieving similar intensities for each elevation. The full measurement sequence takes about 5-15 mins depending on the scattered sunlight intensity. Dark current and offset spectra were measured by blocking incoming light by a mechanical shutter, and were subtracted from the measurement spectra before spectral analysis. In this study, data measured with solar zenith angle (SZA) of less than 75 ° were used to avoid strong influence from the stratosphere (Wang et al., 2014; Wang et al., 2017).

**2.1.2 Spectral analysis**

The MAX-DOAS measured spectra were analyzed using the software QDOAS which is developed by BIRA-IASB (http://uv-vis aeronomie.be/software/QDOAS/). The DOAS fit results are the differential slant column densities (DSCDs), i.e. the difference of the slant column density (SCD) between the off-zenith spectrum and the corresponding zenith reference spectrum. Details of the DOAS fit settings are listed in Table1. A typical DOAS retrieval example for the oxygen dimer ($O_4$),
$NO_2$ and HCHO is shown in Fig. S2. The stratospheric contribution was approximately eliminated by taking the zenith spectra of each scan as reference in the DOAS analysis. Before profile retrieval, DOAS fit results with root mean square (RMS) of residuals larger than 0.003 were filtered.

**2.1.3 HeiPro algorithm description and retrieval parameter settings**

Aerosol and trace gases (i.e., $NO_2$, HCHO) vertical profiles are retrieved from MAX-DOAS measurements using
HeiPro (Heidelberg Profile, developed by IUP Heidelberg) retrieval algorithm (Frieß et al., 2006; Frieß et al., 2011). The inversion algorithm is developed based on the Optical Estimation Method (OEM) (Rodgers, 2000), which employs the radiative transfer model SCIATRAN as the forward model. In general, the maximum a posteriori (MPA) solution $\hat{x}$ is determined by minimizing the cost function $\chi^2$ (Frieß et al., 2006; Frieß et al., 2011; Frieß et al., 2016), which can be express as

$$\chi^2(x) = [y - F(x,b)]^T S_\in^{-1}[y - F(x,b)] + [x - x_a]^T S_a^{-1}[x - x_a] \quad (1)$$

The radiative transfer model or forward model $F(x, b)$ describes the measurement vector $y$ (DSCDs at different elevation angles) as a function of the atmospheric state $x$ (aerosol or trace gas profiles) and meteorological parameter $b$ (i.e. pressure, temperature). The employed atmospheric pressure and temperature profiles were adapted from a climatology database, which contains various monthly and latitudinal dependent trace gases vertical profiles (Clémer et al., 2010; Hendrick et al.,
2014; Wang et al., 2014; Wang et al., 2016).



We assume a fix set of aerosol optical properties with asymmetry parameter of 0.69, a single scattering albedo of 0.90 and ground albedo of 0.05. These values are considered realistic for Shanghai according to (some measurements or previous studies, i.e. Chan et al., 2015). The lowest 4.0 km of the troposphere were divided into 20 layers, each with a thickness of 200 m. A fixed temporal interval of 15 min was used in the inversion. $x_a$ denotes the a priori state vector serving as an additional constraint in the optimization. In order to investigate the impacts of a priori profile shape on the aerosol inversion, four different a priori extinction profiles available in the HeiPro algorithm, i.e. linearly decreasing, exponential decreasing, Boltzmann distribution (smoothed box-shaped) and Gaussian distribution (peaking shape), were used for the sensitivity study (Wang et al., 2016). Details of the sensitivity study are presented in Sect. 3.1. For $NO_2$ and HCHO retrieval, both the a priori profile $x_a$ are exponentially decreasing with scaling height of 3 km, in which the surface concentration of $NO_2$ and HCHO is set to 10.5 and 1.5 ppb, respectively.

The covariance matrixes $S_a$ and $S_\epsilon$ describes the uncertainties and the cross correlation between different layers in the a priori and between measurements at different elevation angles, respectively. Another important quantity is the Jacobian matrix $A=\partial \hat{x}/\partial x$. It represents the sensitivity of the retrieval to the true state. In addition, $A$ provides the degree of freedom of signal (DFS), corresponding to the number of independent pieces contained in the measurement.

## 2.2 Lidar measurements

A polarization backscatter lidar was installed at the same experiment site collocating with the MAX-DOAS instrument. The lidar system is equipped with a diode-pumped Nd:YAG laser emitting laser pulses at 532 and 355 nm by doubling and tripling the laser frequency. The typical pulse energy of the laser is about 20 mJ with a pulse repetition frequency of 20 Hz. The laser beam is emitted with divergence of 0.25 mrad and 158 mm off-axis to the receiving telescope with a field of view of 0.5 mrad, resulting in an overlap height of about 195 m. A constant lidar ratio ($S_p$, extinction to backscatter ratio) of 50 sr was assumed in the lidar retrieval.

Another ozone lidar was applied to detect the $O_3$ profiles at the same time. The differential absorption lidar system emits laser pulse at 316 nm. The typical pulse energy of the laser is about 90 mJ with a pulse repetition frequency of 10 Hz. The laser beam is emitted with divergence of 0.3 mrad and 120 mm off-axis to the receiving telescope with a field of view of 0.5 mrad, resulting in an overlap height of about 300 m.

## 2.3 Ancillary data

Vehicle-based tethered-balloon observations were also preformed regularly at the measurement site during the campaign. The balloon measurement provides information of several meteorological parameters, including temperature, pressure, relative humidity, wind speed, as well as the atmospheric pollutants, i.e. $PM_{2.5}$, $PM_{10}$, $NO_2$ and $O_3$, from ground level up to 900 m above (Li et al., 2015).





In addition, a dynamical/chemical model (WRF-Chem) was used to study the temporal development and the formation of ozone. A detailed description of the model can be found in *Grell et al., 2005*. The simulation domain was set to cover an area of $1200 \times 1200$ km$^2$ ($114°$-$127°$ E, $25°$-$36°$ N) in order to include a number of large cities in the YRD area. The horizontal resolution of the simulation is set to $12 \times 12$ km$^2$ while vertical direction of the model is divided into 26 hybrid
pressure-sigma levels extending from the ground up to 17 km. This setting allows a better reconstruction of the atmospheric status and less impacts due to the diverse emissions of these cities.

## 3 Results and discussion

### 3.1 Dependency of retrieval on a priori profile

The inversion of aerosol extinction profiles was achieved by fitting the O$_4$ DSCD measurements to the forward model
simulations. Previous studies show that there is a systematic uncertainty on the O$_4$ absorption, in which the uncertainty of the O$_4$ absorption was estimated to be ~25% (Clémer et al., 2010; Großmann et al., 2013; Vlemmix et al., 2015). However, reason of the uncertainty is not yet well understood. The uncertainty is typically corrected by multiplying the O$_4$ absorption cross section with a constant correction factor (Wagner et al., 2009; Clémer et al., 2010; Wagner et al., 2011; Chan et al., 2015; Wang et al., 2016). By comparing the measured and modelled O$_4$ absorptions, we estimated that the literature O$_4$
absorption cross section was underestimated by 20%. Therefore, a scaling factor of 1.2 was selected to multiply with the O$_4$ cross section for the O$_4$ retrieval band between 425 and 470 nm.

In order to investigate the impacts of different a priori on aerosol retrieval, a cloud-free day of 17 May 2016 was selected for the aerosol a priori sensitivity study. Fig. 1(a) shows the available configuration of aerosol a priori profile in the HeiPro algorithm, including linear, exponential, Boltzmann and Gaussian shapes. The corresponding retrieved aerosol
extinction profile is shown in the Fig. 1(b) together with lidar observation. The retrieval results using linear, exponential and Boltzmann a priori aerosol profile are very similar displaying the maximum aerosol extinction close to the ground. Aerosol extinction profile retrieved using the Gaussian a priori profile shows the best agreement with simultaneous lidar measurements exhibiting an elevated layer during this period. Furthermore, the diurnal aerosol extinction profiles retrieved with different a priori and the lidar measurements are shown in the right panel of Fig. 1. The retrieval with the Gaussian a
priori profile also shows a better consistency with lidar results during the whole day. The in situ measurements of particle mass concentrations can be also used to semi-quantitative validate the MAX-DOAS retrieval of aerosol extinction coefficients (Wang et al., 2016). We compared the retrieved aerosol extinction profiles to the balloon-based PM$_{2.5}$ measurements. As shown in Fig. S3(a), Aerosol profile retrieved using Gaussian a priori shows the best agreement with the balloon-based measurements, both measurements show a peak value at about 0.75 km above ground level.
Table 2 summarized the parameters of aerosol retrieval performance. The retrieval errors and resulting cost functions using Gaussian a priori are the smallest among all a priori profiles. Moreover, the DFS is about 2.96 when a Gaussian a priori was used. The DFS value suggests at least two independent pieces of information can be determined from the



measurements. So the sensitivity study indicates that the Gaussian a priori profile is the most realistic option for aerosol retrieval during this campaign. As a consequence, Gaussian aerosol profile is selected as the a priori for all aerosol retrieval in this study.

Besides, aerosols strongly influence the effective light path of scattered sun-light in the atmosphere and the Slant Column Densities (SCDs) of trace gases. Therefore, we have examined the sensitivity of trace gas retrieval to aerosol profile. Aerosol profiles retrieved with different a priori profile shapes were used in the differential air mass factor (ΔAMF) calculation for the $NO_2$ profile retrieval in the visible band of 425 to470 nm. $NO_2$ profiles retrieved with different aerosol profiles are shown in Fig. 2(a). The result shows the retrieved vertical distributions of $NO_2$ are easily impacted by the introduced aerosol vertical distributions. The $NO_2$ profile retrieved by the Gaussian aerosol a priori is significantly different from the others. Using balloon-based measurements as a reference, the $NO_2$ profile retrieved with the Gaussian aerosol a priori shows the best agreement. For the other three retrievals, $NO_2$ concentrations at upper layers are significantly lower than the balloon measurement. Moreover, the $NO_2$ profile retrieved using Gaussian aerosol a priori profile as inputs is correlated better (R=0.93) with balloon-based $NO_2$ concentration profiles than others. In Fig. 2(b), the mean difference and standard deviations between the $NO_2$ profile retrieved using Gaussian aerosol a priori profile as inputs and balloon-based measured $NO_2$ profile (26.14% ± 41.34%) is smaller than the other three retrieved $NO_2$ profiles. All these results indicate that aerosol profile scenarios are very important for the trace gas retrieval.

**3.2 Temporal variations of $NO_2$**

Time series of $NO_2$ concentrations profiles were retrieved from MAX-DOAS measurement using the HeiPro algorithm. In order to convert the $NO_2$ SCDs to tropospheric vertical column densities (VCDs), MAX-DOAS retrieved $NO_2$ profiles, lidar aerosol profiles, averaged temperature and pressure profiles measured by in situ instruments on the balloon were introduced as inputs in the radiative transfer model for the $NO_2$ Air Mass Factors (AMFs) calculation. Fig. 3 shows the temporal variations of hourly averaged tropospheric $NO_2$ VCDs and vertical profiles. During the measurement campaign, significant $NO_2$ VCD peaks were observed on 10 and 16 May 2016, respectively. By extracting $NO_2$ concentration at the lowest layer of the retrieval, the averaged ground level $NO_2$ mixing ratio of 12.4 ppb in FengXian area is generally much lower than that in city center of Shanghai (63.3 ppb) (available from http://www.shanghaiair.sinnapp.com/). The sectoral $NO_x$ emissions in FengXian area can be divided into three major types, i.e. transportation, industrial + residential and power generation. These three emission sources contributed 83% (transportation), 15% (industrial + residential) and 2% (power generation) of the total $NO_x$ emissions in FengXian (Chan et al., 2015). The emission inventory indicates that transportation emission plays a dominant role on the local $NO_x$ concentration. 24 hours Air Mass Backward Trajectories (AMBTs) from 8:00 to 17:00 LST (Local Standard Time) at altitude of 500 m over the experimental site were calculated to assess the role of air mass transport during the $NO_2$ episode periods. As shown in Fig. S4, the peak values of tropospheric $NO_2$ VCDs are closely related to the wind direction. Increased $NO_2$ levels mainly occurred during northwesterly or northerly wind conditions, especially during the episodes on 10 and 16 May. Transportations of $NO_2$ can be also observed from OMI



satellite measurements during these episode days. Figure 4 shows the spatial distribution of tropospheric NO$_2$ VCDs from USTC OMI products on 10 and 16 May, respectively. Major industrial areas such as BaoShan, JiaDing, northern part of PuDongXi in Shanghai and a heavy industrial city (i.e., ZhangJiaGang) are located along the backward trajectories during the NO$_2$ episode days. In contrast, lower tropospheric NO$_2$ VCDs were observed during southerly and easterly wind
conditions where the air masses were coming from unpolluted regions and the East China Sea.

The MAX-DOAS NO$_2$ measurements are also used to valid the USTC OMI NO$_2$ product (Liu et al., 2016). MAX-DOAS NO$_2$ VCDs were temporally averaged over the OMI overpass time of 12:00 to 13:00 LST, while the OMI NO$_2$ data are spatially averaged over pixels within 15 km of the experimental site. Previous study shows that a better approximation of trace gas and aerosol profiles for the tropospheric AMFs calculation can significantly improve the OMI NO$_2$ VCDs over
polluted area by 35-40% and bring them closer to the ground based observations (Lin et al., 2014). Therefore, we have re-computed the OMI NO$_2$tropospheric AMFs by using the combined NO$_2$ profiles, in which the lowest 3 km were adopted from the MAX-DOAS NO$_2$ profile retrieval, while NO$_2$ profiles above 3 km were taken from WRF-Chem simulations. Daily tropospheric NO$_2$ VCDs from MAX-DOAS measurement, NASA and USTC OMI product are shown in Fig. 5(a). The temporal trends of MAX-DOAS and USTC OMI data show similar characteristic. However, the MAX-DOAS measurements
are systematically higher than OMI observations by 23% on average. The discrepancy can be explained by the averaging effect over the large OMI pixel which includes the neighboring clean areas. The correlations between MAX-DOAS and two different OMI products are shown in Fig. 5(b). The USTC OMI products agree better with the MAX-DOAS observations with Pearson correlation coefficient (R) of 0.95 (slope of 0.74 and offset of -2.09×10$^{15}$ molec/cm$^2$), while the correlation between MAX-DOAS and NASA OMI product is 0.71 with slope of 0.17 and offset of 2.69×10$^{15}$ molec/cm$^2$. Compared to
MAX-DOAS measurements and USTC OMI product, The NASA OMI NO$_2$ products report much lower NO$_2$ VCDs especially during these two NO$_2$ episode days. The result suggests that adopting local measurement of atmospheric parameters i.e. aerosols and trace gases profiles in AMF calculation could improve the accuracy of satellite VCDs products.

### 3.3 Ozone vertical distribution

Enhanced surface O$_3$ concentrations were found over rural areas of Shanghai compared to the city center (Geng et al.,
2008; Xing et al., 2011). This is probably resulted from the significant contribution of anthropogenic emissions of NO in the city center which consumes ambient ozone through NO-titration (i.e. NO+O$_3$→NO$_2$+O$_2$). In this study, we focused on the formation pathways and the vertical distribution of O$_3$ in rural areas. In order to validate the vertical ozone distribution from lidar measurements, ozone profiles were compared to simultaneous ozone balloon-based measurements. The comparison result shows a good agreement with each other (see Fig. S3(b)-(d)). Time series of ozone vertical distributions measured by
the lidar and surface ozone concentrations measured by the in situ monitor are shown in Fig. 6(a) and (b), respectively. The measurement result shows significant lower ozone concentrations on 15 and 20 May, 2016. Meteorological data shows that the solar irradiance was relatively low and associated with occasional rain on these two days. Decreases of ozone concentration were probably due to lower solar irradiance affecting the photochemical formation of ozone and rainy





condition favouring the wet removal pathway of atmospheric ozone. High ozone concentrations were observed during the
noontime (12:00-14:00) on 16 and 17 May (case 1), as well as all day of 18 May (case 2).

For Case 1 indicated in Fig. 6(a), surface $O_3$ concentrations measured by the in situ monitor correlated well with the lidar observations at low altitudes, particularly for peak ozone values of 75-80 ppb during these two periods. On 16 May, enhanced ozone values not only occurred at surface level, but also found at altitude of about 1.1 km. In contrast, high ozone concentrations were only found at surface level but no extension to the high altitude at the same time period of 17 May. To
further investigate the causes of enhanced $O_3$ levels at upper altitudes on the 16 May, we calculated the $O_3$ fluxes in both horizontal and vertical directions using WRF-Chem simulations (Jiang et al., 2008). The horizontal $O_3$ fluxes reckoned by multiplying the horizontal wind speed with the $O_3$ concentration of the corresponding grid were illustrated in Fig. 7(a) to (d). The vertical flux, shown in Fig. 7(e), is defined as the product of vertical wind speed and $O_3$ concentration at the corresponding layer. Positive values represent upward transportation. To evaluate the accuracy of model simulations, the
simulated ozone concentrations were validated by the ozone lidar measurements. Figure 8 shows the correlation of model simulation and lidar measurement from 0.09-1.5 km. Both datasets agree well with each other with Pearson correlation coefficient (R) of 0.87 (slope of 0.97 and offset of 7.39 ppb). In addition, the root mean square error (RMSE), root mean square error systematic (RMSEs), root mean square error unsystematic (RMSEu) and the index of agreement (d) in different altitudes were analysed to quantify the differences between the measured and simulated values (Willmott, 1981; Geng et al.,
2007). A summary of the statistical analysis is listed in Table 3. The result indicates that the model is able to reproduce the reality. The flux analysis in Fig. 7 shows that horizontal transportations at 900, 1000 and 1300 m were trivial (2.1, 11.7 and 1.1 ppb m$^{-1}$ s$^{-1}$ averaged net values) between 12:00 and 14:00 on 16 May. The result indicates that the horizontal transportations only show negligible effects on the enhanced ozone concentrations at upper altitudes during this time period. Since the vertical wind speed is relatively low (~0.01 m/s, in Fig. 6(c)), the vertical transportations was considered only to
play a minor role. Both the horizontal and vertical transportations were not significant, thus, enhanced ozone levels at upper altitudes were mainly due to local formation.

In case 2, high $O_3$ concentrations were observed from ground surface to higher altitudes on 18 May. Ozone was mainly concentrated at a layer of 0.9 km high from 03:00 to 8:00 LST. Then the ozone layer began to disperse to adjacent layers from 0.5 to 1.3 km. And the ozone concentration gradually increased to more than 115 ppb from 0.5 to 0.9 km after 12:30.
Time series of the planetary boundary layer (PBL) height retrieved from the Mie lidar measurement is shown in Fig. 6(a) (He et al., 2006). It is found that the PBL is relatively stable (at about 0.9 km) from 0:00 to 8:00 am on 18 May. It suggests that the ozone layer was constrained above PBL height at that moment. Afterwards, the PBL height was subsequently rising after sunrise due to the increase of air temperature. Owing to the rise of PBL height and the downward wind, ozone at upper altitudes was gradually mixed and spread throughout the PBL from 9:00 to 12:30. After 12:30, the horizontally averaged net
flux of $O_3$ at 500, 900, 1000 and 1300 m are -1.9, 1.3, -0.44, and 1.1 ppb m$^{-1}$ s$^{-1}$, respectively. The vertical wind speed at different altitudes is extremely low (<0.005 m/s). The above analysis implies that the increased ozone in the PBL after 12:30



were probably related to local formation. More details about the gas chemical analysis during the promoting $O_3$ formation were discussed in Sect. 3.4.

### 3.4 O₃-NO₂-HCHO in vertical

As discussed in Sect. 3.3, two high ozone concentration episodes was mainly locally formed. VOCs are the important precursors for the $O_3$ formation in urban areas (Kleinman et al., 2001; Zhang et al., 2004; Geng et al., 2007). Previous studies show that the formation of surface $O_3$ is mainly under VOC-sensitive regime in Shanghai (Geng et al., 2008). The production of $O_3$ is not only due to the abundance of VOCs, but also related to the reactions with OH radicals and solar irradiance. As HCHO is one of the major VOCs and strongly correlated with peroxy radicals (Sillman, 1995; Duncan et al., 2010), HCHO

measurement results were used as an indicator to represent the total VOCs here. Observations of $NO_x$ and VOCs vertical distribution can provide indispensable information to investigate the ozone formation pathways.

Vertical distributions of $NO_2$ and HCHO were retrieved from the MAX-DOAS observations during the campaign. Fig. 9 shows the time series of $NO_2$, HCHO and $O_3$ vertical distributions during 15 to 20 May. Missing data is due to cloud filtering and/or low signal to noise ratio of the measurements. The distribution patterns of HCHO and $O_3$ above 500 m were very similar. However, $NO_2$ were mainly concentrated below 500 m. Here we focus on 18 May to investigate the causes of

high ozone concentration. As discussed in the Case 2 of Sect. 3.3, it is found that transportation contribution is trivial compared to the total $O_3$ concentration observed on this day. Moreover, we found the higher HCHO concentration (> 8 ppb) occurred prior to the increase of $O_3$ concentration. High HCHO levels during the time are mainly contributed by the oxidation of biogenic emissions of VOCs from plants, i.e., isoprene. Therefore, high abundance of VOCs and relatively

strong radiance contribute to higher formation rates of $O_3$ for Case 2 on 18 May.

### 3.5 Aerosol profile and evolution

Aerosols depolarization profiles were measured by the Mie-elastic backscatter polarization lidar during the campaign. Depolarization ratio is an indicator of the sphericity of aerosols. Low depolarization ratio indicates aerosols are spherical (Burton et al., 2012; Wong et al., 2017). Lidar results show that depolarization ratios of over 70% of measurements during

the campaign are between 0.001 and 0.03 (see Fig. S5). It demonstrates that aerosols in this area were dominated by spherical particles. In general, there are five major spherical particles in the atmosphere, namely black carbon, organic carbon, sulphate, nitrate aerosols and sea salt. Major sources of atmospheric aerosols in FengXian area could be the sea salt aerosols because the measurement site is close to the coast.

Fig. 10(a) shows an enhancement of aerosol extinction from 19:00 to 22:00 on 9 May, while depolarization ratios at 200

m were decreasing during the same period in Fig. 10(b). The growth of extinction coefficient is due to the emission of biofuel/biomass burning in the surrounding agriculture areas (Du et al., 2011). However, the decrease of depolarization ratios is most likely due the secondary processes in the atmosphere as the emission sources do not change rapidly. In Fig. 10(c), the reduction of depolarization ratios is observed prior to the enhancement of aerosol extinction coefficient from 19:00 to 22:00





on 9 of May. Decrease of aerosol depolarization ratio indicates that the aerosols became more spherical during that time. It is

related to the aerosol aging process which accompanied with the mixing between primary and secondary aerosols, as well as the interactions among aerosols, trace gases and moisture in the atmosphere. In addition, $NO_2$ is an important precursor for atmospheric nitrates particles formation (Myoseon and Kamens, 2001; Wang et al., 2017). Correlation between $NO_2$ concentrations and aerosol extinctions from100m to 1000 m above ground level on 9 May is shown in Fig. 11. Strong correlation is observed between ambient $NO_2$ and aerosol (R=0.63) indicating the significant contribution of $NO_2$ to the

secondary aerosol formation. Under high atmospheric ozone conditions, ambient $NO_2$ is oxidized rapidly to form nitrate aerosols in the atmosphere of Shanghai (Du et al., 2011).

## 4 Summary and conclusions

In this paper, we present measurements of $NO_2$ and HCHO vertical profiles using ground based MAX-DOAS, while aerosol and $O_3$ profiles were measured by lidar at Shanghai from 5 to 23 May 2016. Sensitivity study shows that the a priori

profile is playing an important role in the aerosol profiles retrieval. During the period of this campaign, the shapes of aerosol profiles are similar to Gaussian vertical distribution in Shanghai. Accurate aerosol extinction profiles were found to be very important for the retrieval of $NO_2$ and HCHO vertical distribution. Simultaneous measurements of $NO_2$ profiles obtained from balloon-based in situ instrument agrees well with the MAX-DOAS data. In order to validate the OMI $NO_2$ VCDs, the OMI satellite products from USTC and NASA were compared to the ground based MAX-DOAS observations. USTC OMI

data, using corresponding local trace gases profiles for the AMF calculation, present better correlation (R=0.95) than NASA OMI's (R=0.71) with ground based MAX-DOAS measurements. According to the AMBTs analysis and the spatial distribution of averaged OMI tropospheric $NO_2$ VCDs, the $NO_2$ pollution at FengXian were mainly influenced by transportations from industrial areas located in the north and northwest of Shanghai (BaoShan and JiaDing) and south of Jiangsu province (e.g. ZhangJiaGang).

$O_3$ vertical profiles were measured by lidar. Based on the analysis of horizontal and vertical fluxes of ozone at different altitudes, we know that transportation is not a major influencing factor causing the increase of $O_3$. Similar vertical distributions of HCHO and $O_3$ indicate the local formation was the dominant ozone source during the time. Moreover, secondary aerosol formation process was found based on the analysis of aerosol extinction coefficient and depolarization ratios. A positive correlation between $NO_2$ and aerosols during the campaign indicates the significant contribution of $NO_2$ to

total aerosols in the atmosphere.

## Acknowledgements

This research was supported by grants from National Key Project of MOST (2016YFC0203302), National Natural Science Foundation of China (41575021, 91544212, 41405117) and the Key Project of CAS (KJZD-EW-TZ-G06-01). We





acknowledge the NOAA Air Resources Laboratory (ARL) for making the HYSPLIT transport and dispersion model
available on the Internet (http://ready.arl.noaa.gov/). We thank Shanghai Environment Monitoring Center, Nanjing
University and East China University of Science and Technology of contributing to the balloon-based measurements. We
would like also to thank Hefei Institute of Physical Science, Chinese Academy of Sciences for the technical support of lidar
measurement.

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



**Table 1. Setting for the O₄, NO₂, and HCHO DOAS spectral analysis**

| Parameter | Data source | Fitting interval | | |
|---|---|---|---|---|
| | | O$_4$ | NO$_2$ | HCHO |
| Wavelength range | | 425-490nm | 425-490nm | 336.5-359nm |
| NO$_2$ | 298K, I$_0$* correction (SCD of $10^{17}$ molecules/cm$^2$), Vandaele et al. (1998) | √ | √ | √ |
| NO$_2$ | 220K, I$_0$ correction (SCD of $10^{17}$ molecules/cm$^2$), Pre-orthogonalized Vandaeele et al. (1998) | √ | √ | × |
| O$_3$ | 223K, I$_0$ correction (SCD of $10^{20}$ molecules/cm$^2$), Serdyuchenko et al. (2014) | √ | √ | √ |
| O$_3$ | 243K, I$_0$ correction (SCD of $10^{20}$ molecules/cm$^2$), Pre-orthogonalized Serdyuchenko et al. (2014) | × | √ | √ |
| O$_4$ | 293K, Thalman and Volkamer (2013) | √ | √ | √ |
| HCHO | 297K, Meller and Moortgat (2000) | × | √ | × |
| BrO | 223K, Fleischmann et al. (2004) | × | × | √ |
| H$_2$O | 296K, HITEMP, Rothman et al. (2010) | √ | √ | × |
| Ring | Calculated with QDOAS | √ | √ | √ |
| Polynomial degree | | Order 5 | Order 5 | Order 5 |
| Intensity offset | | Constant | Constant | Constant |

*Solar I$_0$ correction, Aliwell et al., 2002



**Table 2. Cost function, DFS and retrieved errors using different a priori profile in the aerosol retrieval**

| The shape of a priori | Chi square | DFS | Retrieved error (<5%) | Smooth error (<5%) | Noise error (<5%) |
|---|---|---|---|---|---|
| Linearly shape | 48.534512 | 2.957806 | 65% | 65% | 100% |
| Exponential shape | 22.907515 | 2.9162457 | 75% | 75% | 100% |
| Boltzmann shape | 28.533862 | 3.0331712 | 65% | 65% | 80% |
| Gaussian shape | 2.9297998 | 2.9614936 | 100% | 100% | 100% |



**Table 3. Statistical analysis for the simulation of wind speed and ozone concentrations in different altitudes**

|              | Altitudes(m) | RMSE | RMSEs | RMSEu | d |
|--------------|--------------|------|-------|-------|------|
| Wind         | 580          | 0.61 | 0.31  | 0.49  | 0.83 |
| Speed        | 670          | 0.58 | 0.28  | 0.53  | 0.69 |
| $(m.s^{-1})$ | 800          | 0.70 | 0.40  | 0.59  | 0.79 |
| Ozone        | 580          | 6.6  | 3.2   | 6.5   | 0.63 |
|              | 670          | 8.1  | 4.1   | 7.0   | 0.57 |
| (ppbv)       | 800          | 7.3  | 3.7   | 6.7   | 0.59 |




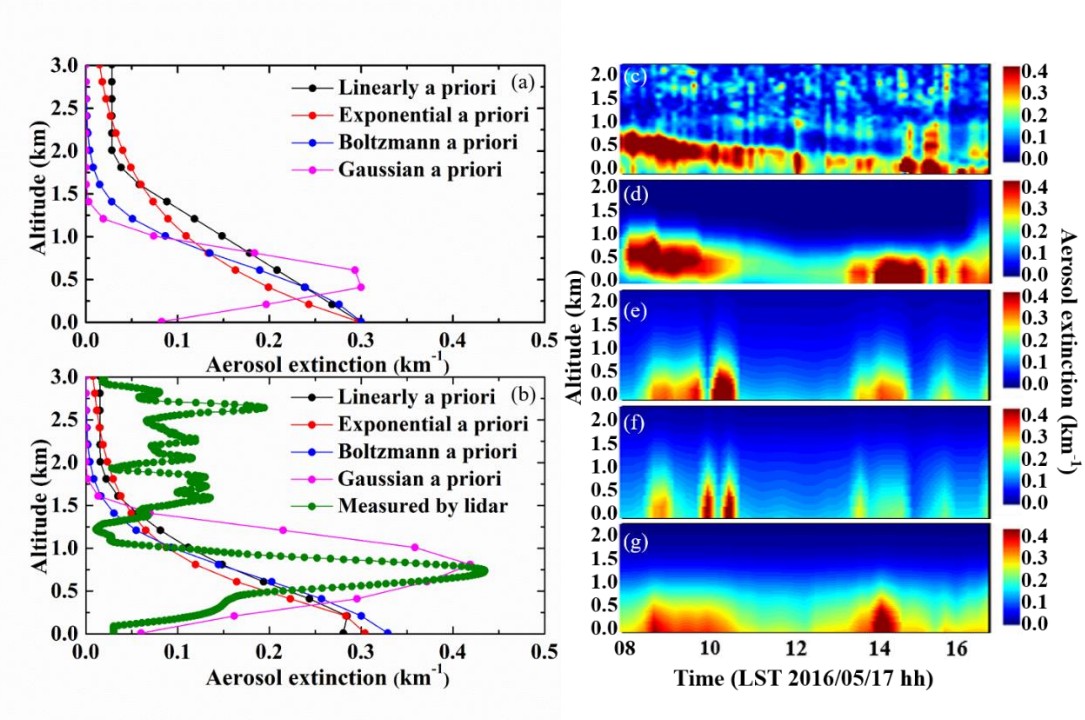

**Figure 1. Different aerosol extinction a priori and corresponding retrieval on 17 May 2016. (a) different a priori aerosol extinction profiles used in HeiPro. (b) shows four kinds of retrieved aerosol profiles and the lidar measured profile at 8:30. Diurnal aerosol extinction coefficient at 477 nm from (c) Mie-Scattering polarization lidar and retrieved using different a priori (d) Gaussian, (e) exponential, (f) linear and (g) Boltzmann.**







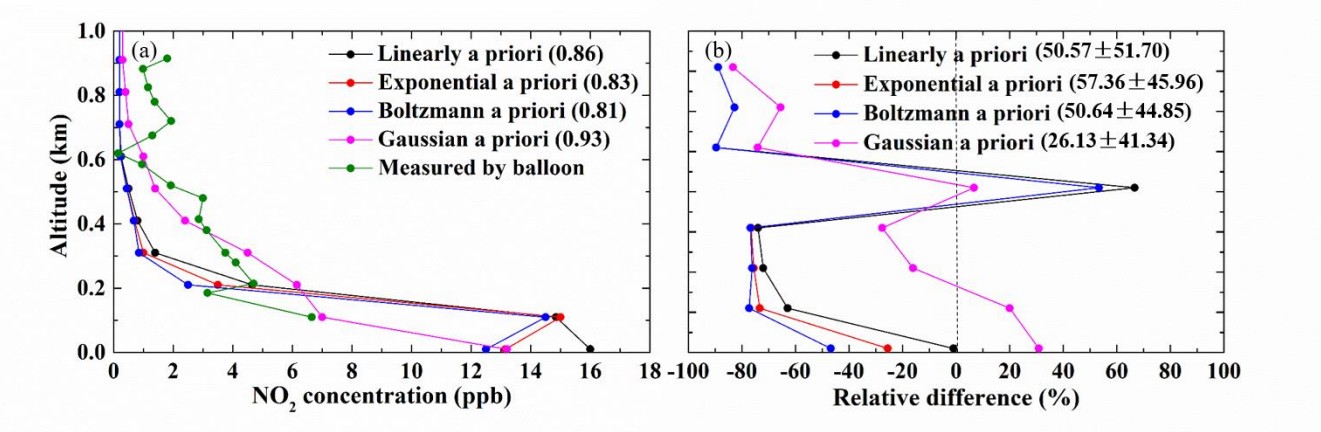

Figure 2. Comparison of $NO_2$ profiles retrieved using four different aerosol a priori profiles as inputs and measured by tethered-balloon. (a) shows the four retrieved vertical $NO_2$ concentrations and balloon-based measurement, as well as the correlation coefficients. (b) shows the mean differences and standard deviations between four different retrieved $NO_2$ profiles and the balloon-based $NO_2$ profiles.



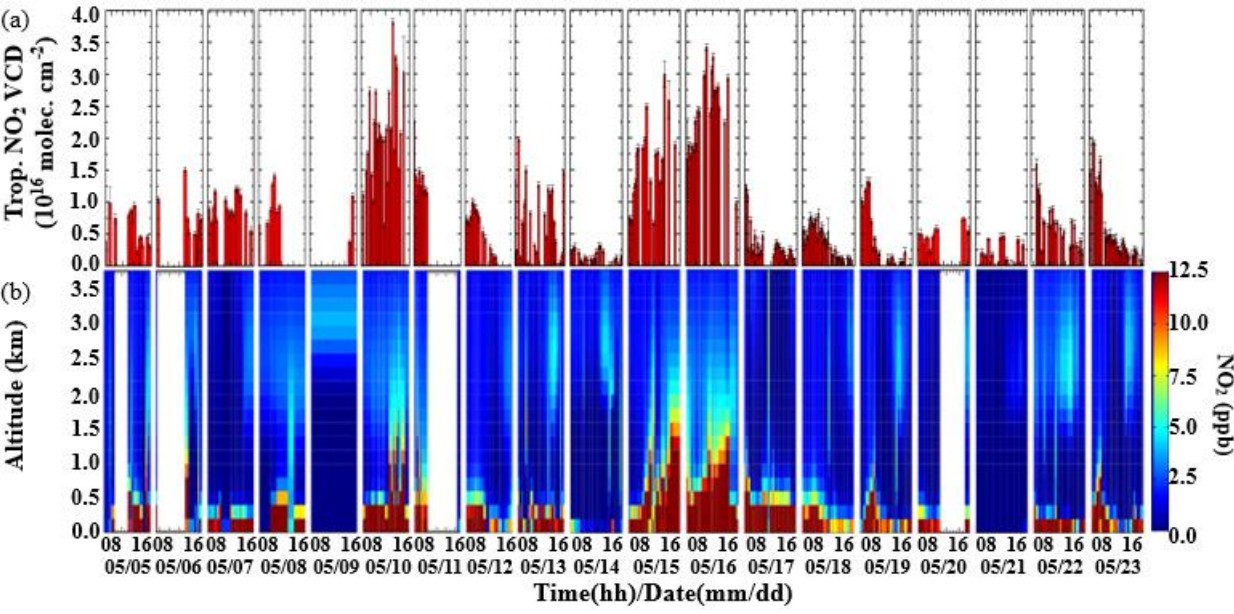

**Figure 3. Time series of hourly averaged (a) NO₂ VCDs and (b) NO₂ vertical profiles from MAX-DOAS measurements.**








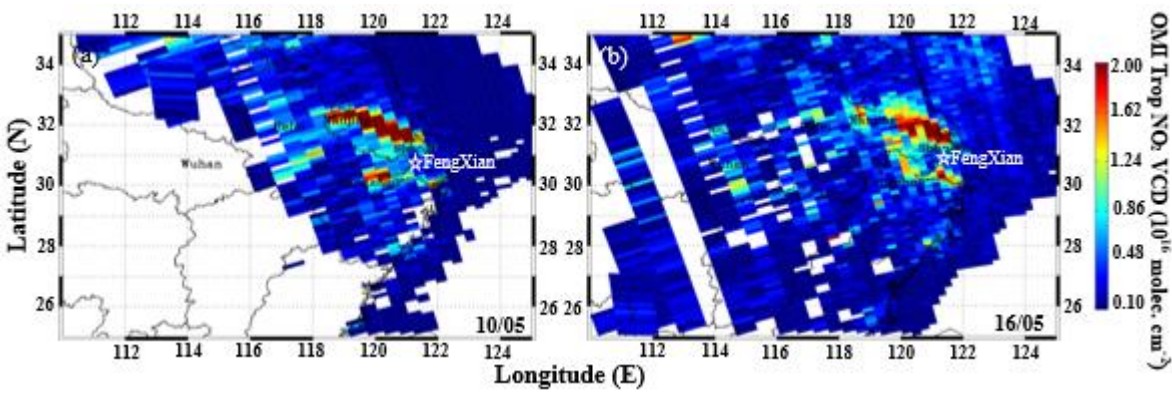


**Figure 4. Spatial distribution of OMI tropospheric NO₂ VCDs on (a) 10 and (b) 16 May, 2016.**








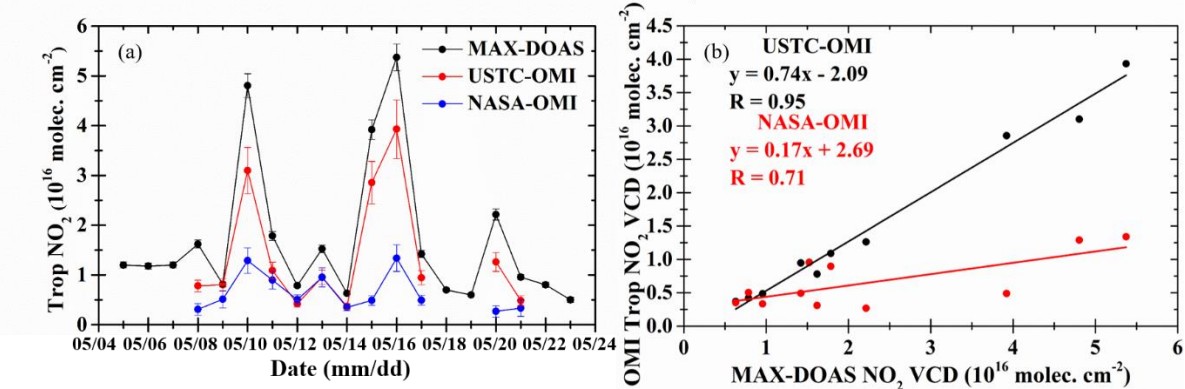

**Figure 5. Comparison of tropospheric NO₂ VCDs between ground based MAX-DOAS measurement and OMI satellite observation. (a) shows time series of daily averaged tropospheric NO₂ VCDs. MAX-DOAS data are averaged around the OMI overpass time. (b) shows the correlation of daily averaged tropospheric NO₂ VCDs measured by MAX-DOAS with USTC OMI and NASA OMI satellite data. The OMI measurements are spatially averaged over the grid cells within 15 km of ground location around the campaign site.**





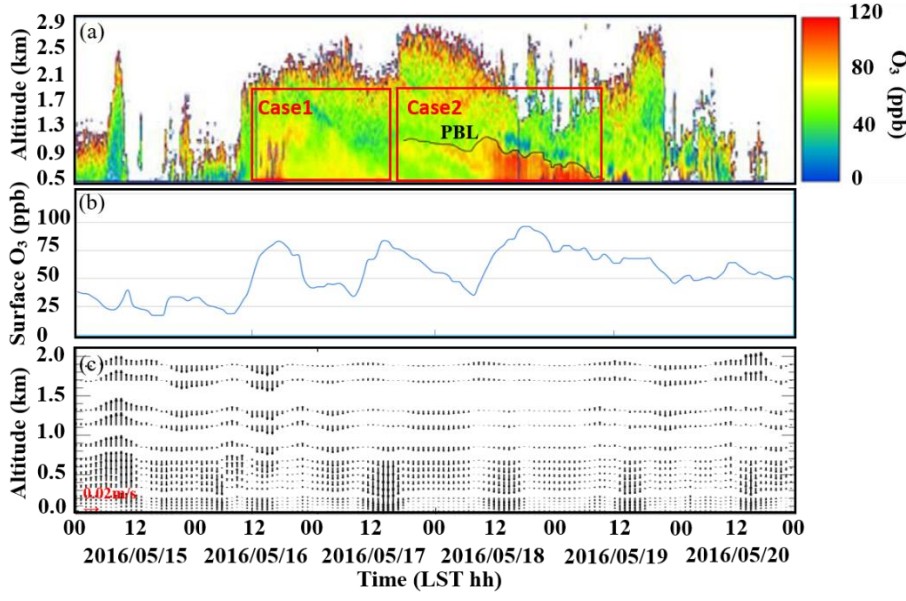

**Figure 6. Time series of (a) ozone vertical distributions measured by ozone lidar, (b) surface O₃ concentrations detected from in situ ozone instrument and (c) vertical wind profiles simulated by WRF in FengXian from 15 to 20 May, 2016.**









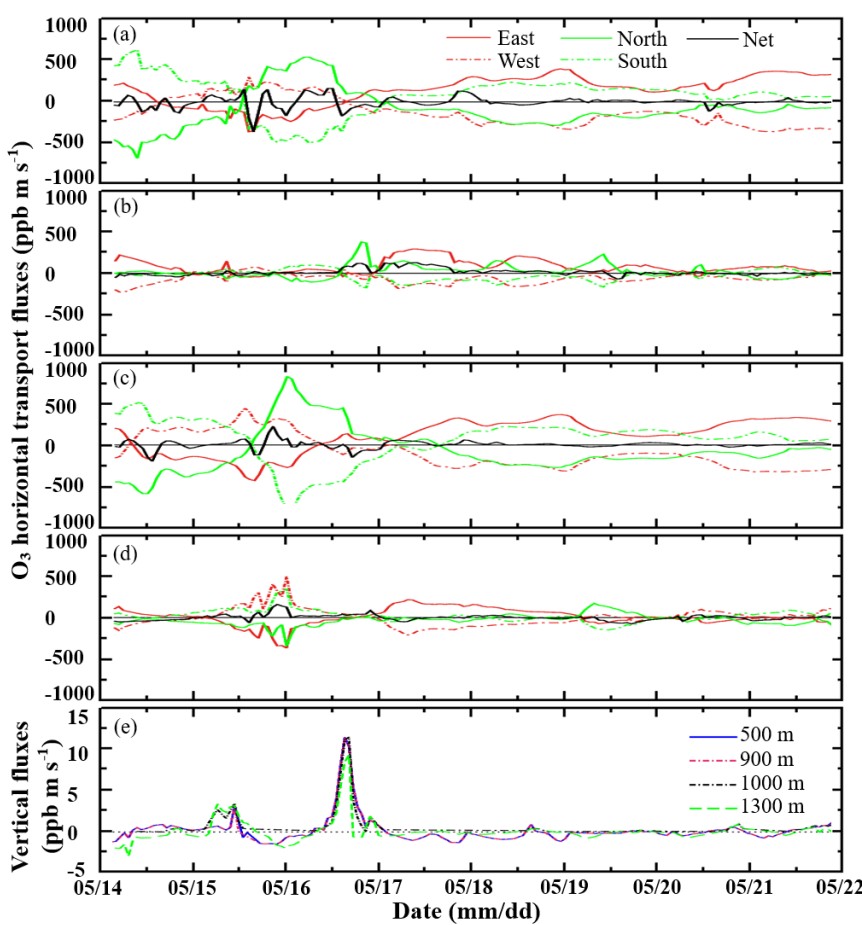

**Figure 7. Ozone horizontal transport fluxes at (a) 500 m, (b) 900 m, (c) 1000 m and (d) 1300 m, as well as (e) vertical transport fluxes.**



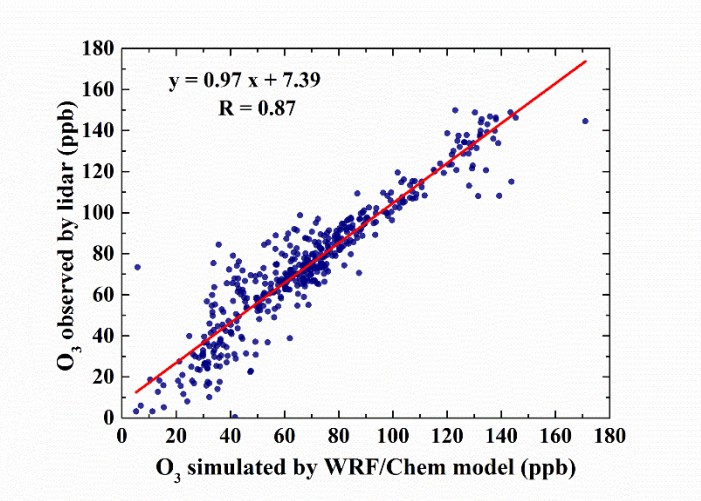

**Figure 8. Correlation of O₃ concentration at different altitudes between WRF-Chem simulation and lidar measurement.**





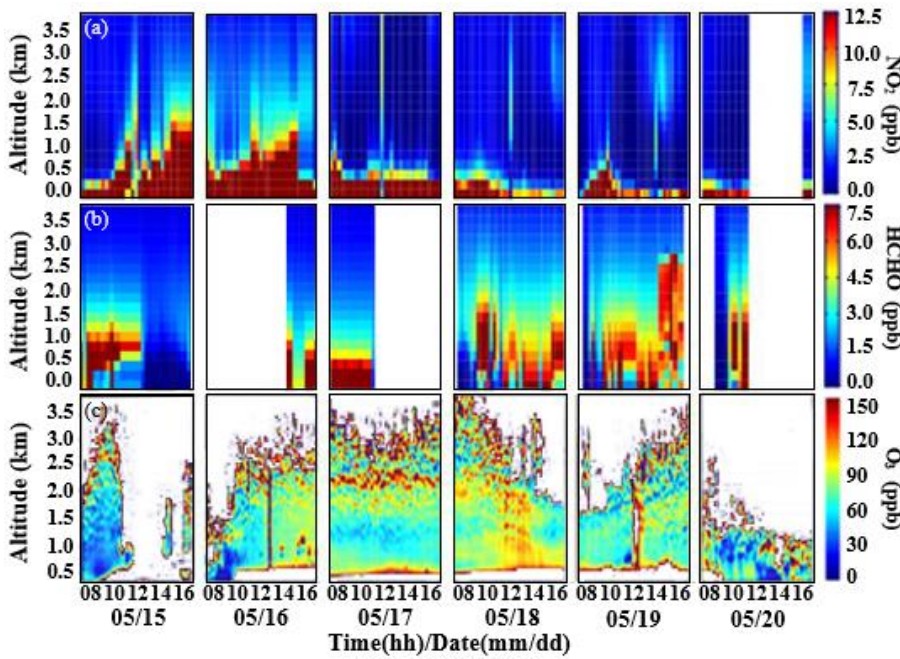


**Figure 9. Time series of retrieved (a) NO₂ and (b) HCHO vertical profiles from MAX-DOAS, as well as (c) O₃ vertical profiles measured by lidar.**






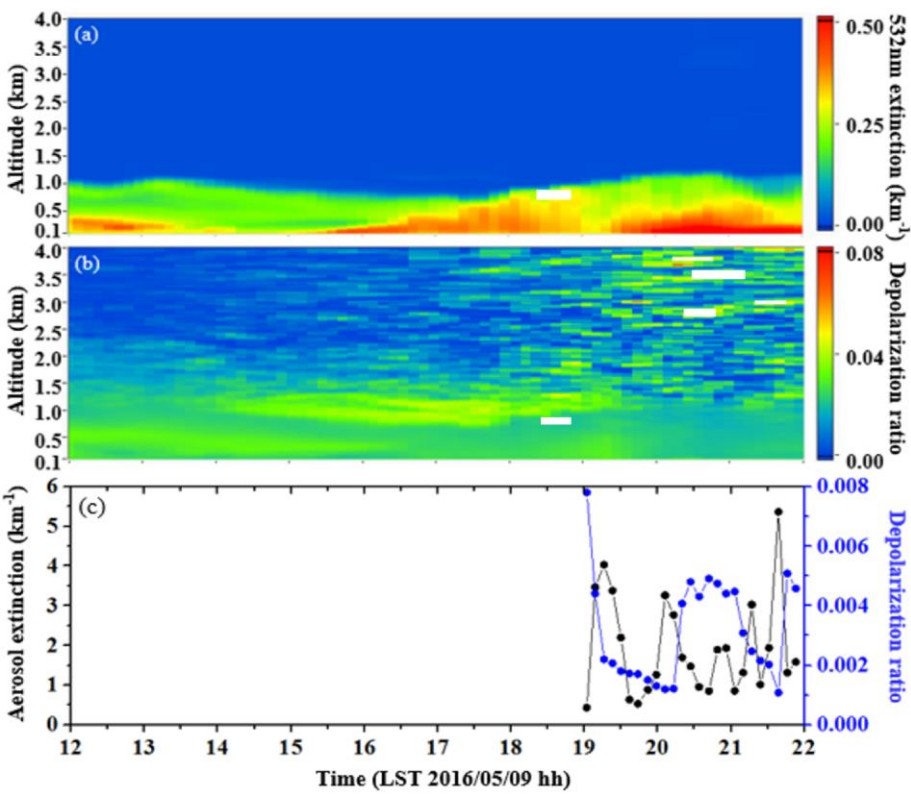


**Figure 10. Aerosol extinction coefficient (a) and depolarization ratio (b) on 9 May. (c) shows time series of aerosol extinction and depolarization ratios at 200 m from 19:00 to 22:00 on 9 May, 2016.**








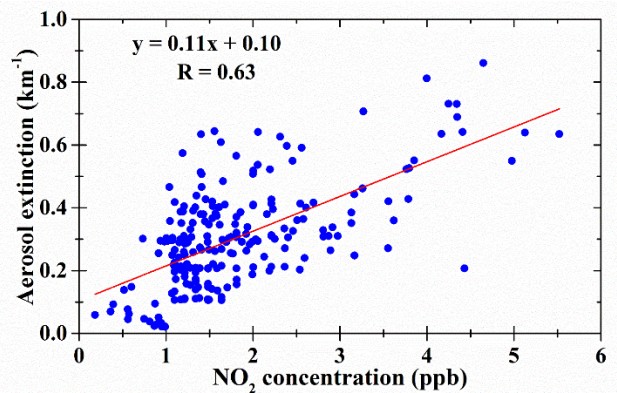

**Figure 11. Scatter plots of aerosol extinction coefficient measured by Mie-Scattering depolarization lidar versus**
**MAX-DOAS measurement of NO₂ from 100 to 1000 m above ground level.**