# Peer review of "Observations of the summertime atmospheric pollutants vertical distributions and the corresponding ozone production in Shanghai, China"

_Atmospheric Chemistry and Physics, 2017_

## Referee Comment (RC1) · Anonymous Referee #1 · 14 Sep 2017

The manuscript investigated the summertime distributions of air pollutants obtained by MAX-DOAS and lidar measurements, especially for ozone and its precursors. In this study, sensitivity study of different a priori shapes to the MAX-DOAS aerosol profile retrieval were carried out, which shows significant influences on the aerosol and trace gases profile retrieval. Moreover, the MAX-DOAS measured tropospheric NO2 VCDs show a good agreement with OMI satellite observations, indicating that the accuracy of satellite products over polluted area can be improvement by introducing local atmospheric parameters in the AMF simulation. It was suitable and meaningful for the ACP community. However, there still some concerns need to be addressed before it can be accepted for publication on ACP journal.

- 1. Section 2.1, Line 109: what's kind of influence from the stratosphere? Please explain clearly.
- 2. Section 2.1 mentioned the lowest elevation angle is 3 degree. Please clarify why these angles were chosen, as lower elevation angles are more sensitive to the near ground aerosol and trace gases.
- 3. In section 2.1 the author mentioned that the measurement sequence takes 5-15 minutes, however, section 2.1.3 wrote that a fix temporal resolution of 15 minutes is used. Did the author average the DSCDs if more that one sequence were measured within 15 minutes? Please describe the procedure in more detail.
- 4. Section 2.3, please provide the information of the measurement instruments use for PM 2.5, PM10, NO2 and O3 measurements on the balloon.
- 5. Section 3.1, the sensitivity study indicates that the Gaussian a priori profile is the most realistic option for aerosol retrieval during this campaign. Could the authors explain more about the possible reasons for this vertical aerosol distribution? Line 190: Is Table 2 the summary of the aerosol retrieval performance or just a case?
- 6. Section 3.3: There a peak of ozone vertical fluxes around May 16 noontime can be observed in Fig. 7. However, the authors have concluded the ozone concentration is less impacted by the vertical and horizontal transportation (line 279-280). Are they contradictory?

Technique corrections:

The quality of writing and style of English should be improved in general. A careful check over the entire manuscript about the typing errors, use of the notations, citations, etc, is also required. A few examples of the technical errors are corrected as the followings:

There are lack of much space in the manuscript, e.g. Line 32: add a space between "NO2" and "Vertical". Please read the whole manuscript carefully and make the corrections.

Line 75: Lidar→lidar

Line 105: calculated depending on  $\rightarrow$  adjusted according to

Line 107: by a mechanical shutter  $\rightarrow$  using a ...

Line 108: delete "of"

Line 240: The discrepancy  $\rightarrow$  These discrepancies

Line 245: The NASA→the NASA

Line 283: 8:00→08:00

Fig. 4: please indicate the date using the unify format in the whole manuscript, e.g. mm/dd

---

## Referee Comment (RC2) · Anonymous Referee #2 · 17 Sep 2017

General comments

This paper presents a comprehensive measured dataset of vertical profiles of aerosol extinction, NO2, O3 and HCHO at a rural site in the area of Yangtze River Delta (YRD) with well established MaxDOAS and lidar techniques in their research groups, respectively. The successful deployment of these new measurement techniques in Chinese megacity areas will potentially open up a broader view (especially over the vertical dimension) on the exploration of the air pollution formation mechanism. The retrieved NO2 concentrations from the MaxDOAS instrument is assisted by the lidar observed

aerosol extinctions and further validated by the ballon measured NO2 concentrations. The well established MaxDOAS NO2 concentrations were then used to validate the satellite meausrement results over YRD. The lidar measured O3 is compared with the chemical transport model results and showed reasonable good agreement. Overall, the authors presented a number of high quality vertical profiles of aerosol extinction, NO2, O3 and HCHO in the rural area of YRD for the first time and is an important and valuable contribution to the community of atmospheric chemistry which definitely worth publication in the journal of ACP. Nevertheless, some parts of the paper still need some further modificiation or polishment. For that purpose, I had the following comments for the consideration of the authors.

Specific comments

1. Figure 1 and related text, is the choice of Gaussian a priori really the best one? The selected lidar measurement were performed at 08:30 when the Plantery boundary layer was not fully developed. So the high aerosol concentrations appeared at around 500 m. Nevertheless, I assume the aerosol profile will be different for 12:00. And even for the lidar results of 08:30, the Gaussian a priori profile seems to be significantly underestimated the observed aerosol concentrations above 1 km. In this case, may be a Lorentz a priori would be better.

2. Figure 2 and related text, what was the time of the ballon measurement performed? The ballon measurement may take some time from lower to higher alitiude so that the observed NO2 profile is composed by both the vertical and temperal change of the probed air masses. Which instrument is used for the measurement of NO2? As we know, the NO2 measured by the Mo-CLD instrument will be strongly influenced by the NOz in the atmosphere. Likewise, I suggest the authors to have more detailed description of the experimental part of the Vehicle-based tethered-balloon observations of PM and O3.

3. Figure 5, if the NASA OMI NO2 products is biased in such a way over the Chinese

megacities, it is an important message to the community which needs more highlight in discussion and conclusion. Is the NASA OMI NO2 products also strongly biased on the retrieval of the spatial pattern of NO2 compared to that of the USTC OMI NO2 product (shown in Figure 4)?

4. Figure 6-9 and related text (mainly Sect. 3.3 and 3.4), the authors have used WRF-chem simulated results to assist the observed O3 vertical profiles for the diagnosis of the ozone sources. And it is argued the O3 high values observed is locally formed which is probably related to the high VOCs and relatively strong solar irradiance. The currently analysis is strongly depending on the observational deduction. What does the WRF-Chem model tell? I encourage the authors to abstract more quantitative results from the used WRF-Chem simulations.

5. Line 308 -309, "High HCHO levels during the time are mainly contributed by the oxidation of biogenic emissions of VOCs from plants, i.e., isoprene." Do the authors have evidences for such statement? At least a reference is needed for this statement.

6. Line 317 – 318, "Major sources of atmospheric aerosols in FengXian area could be the sea salt aerosols because the measurement site is close to the coast." This is probably not true. In China, even for the area close to the coast, the aerosols were normally dominated by sulfate, nitrate, amonia and organics. I suggest the authors to look up the avaiable published results of aerosol chemical composition in Shanghai and sourrouding areas.

7. The judgement of the secondary aerosol formation from a correlation of NO2 and aerosol extinction coefficient seems to be quite arbitrary. And a R = 0.63 can not be well recognized as a significant correlation. More discussions are required to support this deduction. And the information by Figure 11 is very limited, I suggest to move this figure into supporting materials.

Technical comments:

Line 168, change 'Dependency' to be 'Dependence'

Line 227, USTC shall be explained when appeared for the first time.

Line 580, Figure caption of Figure 2, (a) . . . as well as the correlation coefficients shown in the brackets. (b) . . . shows the mean differences and standard deviations in the brackets . . .

Line 626, Spatial distribution of USTC OMI tropospheric NO2 . . .

---

## Author Response (AR1)

**Point-to-point responses**

We appreciate the reviewers for their valuable and constructive comments, which are very helpful for the improvement of the manuscript. We have revised the manuscript carefully according to the reviewers' comments. We have addressed the reviewers' comments on a point-to-point basis as below for consideration, where the reviewers' comments are cited in black, and the responses are in blue.

**Referee #1**

General comments
(1) Section 2.1, Line 109: what's kind of influence from the stratosphere? Please explain clearly.
R: In twilight geometry (mostly SZA > 75°), the scattering mainly takes place in the lower stratosphere and upper troposphere, zenith-sky DOAS measurements are very sensitivity to stratospheric absorbers, while the sensitivity to absorbers close to the surface is relatively lower. In other words, absorbers in stratosphere contribute significantly to the measurements, especially for lower elevation angles during early morning and late evening. In this study, we are focusing on the tropospheric absorbers close to the ground surface and therefore have filtered out the measurements with SZA > 75°.

(2) Section 2.1 mentioned the lowest elevation angle is 3 degree. Please clarify why these angles were chosen, as lower elevation angles are more sensitive to the near ground aerosol and trace gases.
R: We agree with the reviewer that lower elevation angles are more sensitive to the near ground aerosols and trace gases. However, the viewing geometry was limited by the complex topography, i.e., tall buildings and hills surrounding the measurement site. Therefore, the lowest elevation that we could reach is 3°. We have now clarified this issue in the manuscript Line 104 to 106.

(3) In section 2.1 the author mentioned that the measurement sequence takes 5-15 minutes, however, section 2.1.3 wrote that a fix temporal resolution of 15 minutes is used. Did the author average the DSCDs if more than one sequence were measured within 15 minutes? Please describe the procedure in more detail.
R: A full measurements sequence takes about 5-15 minutes depending on the scattered sunlight intensity for our MAX-DOAS instrument. For the vertical profiles retrieval of aerosol and trace gases, DSCDs or relative intensities measured at several elevation angles, relative to a zenith sky spectrum of the same sequence, serve as input measurement vector. In practical, all the individual information (DSCDs or intensities) within the fixed setting time interval (e.g. 15 min) was utilized as corresponding element of the measured vector in each iteration process of HeiPro algorithm. If there

are more than one scan during the time interval, all the measured vectors will be included in the retrieval processes without averaging or screening. Please refer to Line 136-137.

(4) Section 2.3, please provide the information of the measurement instruments use for PM $_{2.5}$, PM10, NO2 and O3 measurements on the balloon.

R: The $O_3$ concentrations were measured by a UV photometric analyzer with the detection limit of 0.5 ppb (Thermo 49i, Thermo Fisher, USA). $NO_2$ concentrations were measured using a nitrogen oxides analyzer with the detection limit of 0.4 ppb (Thermo 42i, Thermo Fisher, USA). Moreover, particulate matters ($PM_{2.5}$ and $PM_{10}$) concentrations were measured with the tapered element oscillating microbalance (TEOM) particle monitors (Thermo TEOM 1405, Thermo Fisher, USA), which sampled air at a flow rate of 10 L/min. Separation of particle size is realized by using an impactor with slit width of 47 nm. The information of measurement instruments were also provided in the Sect. 2.3 of the revised manuscript. Please refer to Line 163-166.

(5) Section 3.1, the sensitivity study indicates that the Gaussian a priori profile is the most realistic option for aerosol retrieval during this campaign. Could the authors explain more about the possible reasons for this vertical aerosol distribution?

Line 190: Is Table 2 the summary of the aerosol retrieval performance or just a case?

R: Yes. Both aerosol profiles observed by the Mie-Scattering depolarization lidar and vertical $PM_{2.5}$ concentrations measured by in-situ instrument mounted on the balloon show that particulate matters were distributed as Gaussian shape in vertical direction. It means that the true vertical distribution of aerosol is indeed Gaussian shape during the campaign. The reasons for the Gaussian distribution of aerosol could be the influences of the boundary layer evolution. For example, fig. 1(b) presents that high aerosol concentrations mainly appeared at 750 m at 08:30 from the lidar measurements and the boundary layer was not fully developed at that moment.

Table 2 is the summary of the aerosol retrieval performance on May 17, we also have clarified in the manuscript (Line 196).

(6) Section 3.3: There a peak of ozone vertical fluxes around May 16 noontime can be observed in Fig. 7. However, the authors have concluded the ozone concentration is less impacted by the vertical and horizontal transportation (line 279-280). Are they contradictory?

R: They are not contradictory. The high concentration $O_3$ was observed at about 900 meters around 12:15 on May 16 in Fig. 6, while the peak of ozone vertical fluxes appeared around 14:00 on May 16 shown in Fig. 7. The ozone vertical flux was 1.8 ppb m s$^{-1}$ at 12:15. Moreover, the lidar results shows that the ozone concentrations at higher (i.e. 1000 m) or lower (800 m) altitudes, as well as the surface concentration were just one-half of the peak value at 900 m. The vertical gradient of ozone concentration can also indicate the ozone concentration is less impacted by the vertical transportation.

Technique corrections

①There are lack of much space in the manuscript, e.g. Line 32: add a space bet "NO2" and "Vertical". Please read the whole manuscript carefully and make the corrections.

②Line 75: Lidar→lidar

③Line 105: calculated depending on→ adjusted according to

④Line 107: by a mechanical shutter→using a …

⑤Line 108: delete "of"

⑥Line 240: The discrepancy→These discrepancies

⑦Line 245: The NASA→he NASA

⑧Line 283: 8:00→08:00

⑨Fig. 4: please indicate the date using the unify format in the whole manuscript, e.g. mm/dd

R: We have following these suggestions and corrected these mistakes accordingly.

**Referee #2**

General comments

(1) Figure 1 and related text, is the choice of Gaussian a priori really the best one? The selected lidar measurement were performed at 08:30 when the Plantery boundary layer was not fully developed. So the high aerosol concentrations appeared at around 500 m. Nevertheless, I assume the aerosol profile will be different for 12:00. And even for the lidar results of 08:30, the Gaussian a priori profile seems to be significantly underestimated the observed aerosol concentrations above 1 km. In this case, may be a Lorentz a priori would be better.

R: In the manuscript, we only tested the available configuration of aerosol a priori profile in the HeiPro algorithm, including linear, exponential, Boltzmann and Gaussian shapes. Diurnal aerosol extinction coefficient at 477 nm retrieved using Gaussian, exponential, linear and Boltzmann a priori were performed in Figure 1(d)-(g), respectively. Aerosol vertical profiles retrieved using Gaussian a priori show the best agreement with lidar measurement results for the whole day (between Figure1 (c) and (d)). However, the aerosol profile retrieved using Gaussian a priori seems to be significantly underestimated aerosol above 1 km comparing to lidar measurement results. As the MAX-DOAS measurement is not sensitive to aerosol at higher altitude, therefore, it tends to follow the a priori profile if the aerosol extinction is set to some small values.

The Lorentz a priori is not available in HeiPro. Here we defined a Lorentz a priori with equivalent aerosol loading compared to the other four a priori profiles in Figure R1(a). Then, the Lorentz a priori was introduced in the aerosol retrieval as shown in Figure

R1(b).

[Figure]

**Figure R1. Different aerosol extinction a priori and corresponding retrieval on 17 May 2016. (a) different a priori aerosol extinction profiles used in HeiPro. (b) shows four kinds of retrieved aerosol profiles and the lidar measured profile at 8:30.**

We found that the aerosol profile retrieved from Lorentz a priori shows a significant overestimation compared to the lidar results between 1 and 2 km. Besides, the degree of freedom of signal (DFS) of using Gaussian a priori (2.96) is higher than that of using Lorentz a priori (2.4). The retrieved error of adopting Lorentz a priori is relatively higher than the Gaussian a priori at the lowest 1 km (see Figure R2). In general, the Lorentz a priori shows the potential to retrieve the reasonable results. Maybe we leave it for the upcoming algorithm development.

[Figure]

**Figure R2. The retrieved error of using Gaussian and Lorentz a priori for aerosol retrieval.**

(2) Figure 2 and related text, what was the time of the balloon measurement performed? The balloon measurement may take some time from lower to higher altitude so that the observed $NO_2$ profile is composed by both the vertical and temporal change of the probed air masses. Which instrument is used for the measurement of $NO_2$? As we know, the $NO_2$ measured by the Mo-CLD instrument will be strongly influenced by the $NO_2$ in the atmosphere. Likewise, I suggest the authors to have more detailed description of the experimental part of the Vehicle-based tethered-balloon observations of PM and $O_3$.
R: A nitrogen oxides analyzer (Thermo 42i, Thermo Fisher, USA) was mounted on the balloon to measure $NO_2$ concentration during the balloon moved upward. The measurement was performed from 08:20 to 08:40 in the morning. It means the balloon takes about 20 minutes to ascend from ground up to 900 m. We used a fixed time

interval of 15 minutes for the input vector in HeiPro during the retrieval of $NO_2$ vertical profiles. The $NO_2$ profile depicted in Figure 2 was retrieved from period between 08:25 and 08:40.

The concentrations of $O_3$ were measured by a UV photometric $O_3$ analyzer (Thermo 49i, Thermo Fisher, USA). Particulate matters ($PM_{2.5}$ and $PM_{10}$) were measured with a tapered element oscillating microbalance (TEOM) particle monitors (Thermo TEOM 1405, Thermo Fisher, USA), which is sampling with a flow rate of 10 L/min. Separation of particle size is realized by using an impactor with slit width of 47 nm. Moreover, the balloon was launched to ascend at a steady speed of 0.5m/s until a maximum height of 1000m.

We have also supplemented this information in Line 163-166 in the manuscript.

(3) Figure 5, if the NASA OMI $NO_2$ products is biased in such a way over the Chinese megacities, it is an important message to the community which needs more highlight in discussion and conclusion. Is the NASA OMI $NO_2$ products also strongly biased on the retrieval of the spatial pattern of $NO_2$ compared to that of the USTC OMI $NO_2$ product (shown in Figure 4)?

R: As described in the manuscript, the NASA OMI $NO_2$ products report lower $NO_2$ VCDs especially during $NO_2$ episode days. The improvement of USTC OMI $NO_2$ products is mainly related to the usage of better $NO_2$ and aerosol vertical profiles for the AMF calculation. We also include the important message in the discussion and conclusion part. Please refer to Line 252 to 255 and Line 357 to 358.

The spatial distribution of USTC and NASA OMI $NO_2$ products are shown in Figure 3R and 4R, respectively. With same screening criteria (relative error of $NO_2$ VCDs $\leq$ 50%), the USTC product shows better coverage than NASA's. Moreover, considerable underestimation over the hotspots can be observed during the $NO_2$ episodes in the NASA's product. Transportation track of air mass containing $NO_2$ is also more obvious in USTC products (Figure R3) than NASA products (Figure R4).

[Figure]

**Figure R3. Spatial distribution of USTC OMI tropospheric $NO_2$ VCDs on (a) 10 and (b) 16 May, 2016**.

[Figure]

**Figure R4. Same as Figure R3 using NASA OMI tropospheric NO₂ VCDs**.

(4) Figure 6-9 and related text (mainly Sect. 3.3 and 3.4), the authors have used WRF Chem simulated results to assist the observed O3 vertical profiles for the diagnosis of the ozone sources. And it is argued the O3 high values observed is locally formed which is probably related to the high VOCs and relatively strong solar irradiance. The currently analysis is strongly depending on the observational deduction. What does the WRF-Chem model tell? I encourage the authors to abstract more quantitative results from the used WRF-Chem simulations.

R: We used WRF-Chem model to simulate the $O_3$ concentration and further to calculate $O_3$ horizontal and vertical fluxes. The $O_3$ concentration simulated by WRF-Chem and measured by lidar shows good agreement (R = 0.87). The horizontal averaged net flux of $O_3$ at 500, 900, 1000 and 1300 m are -1.9, 1.3, -0.44 and 1.1 ppb m s$^{-1}$ after 12:30 on 18 May, respectively. Moreover, the averaged vertical flux of $O_3$ at 500, 900, 1000 and 1300 m are -1.2, -1.2, -0.08 and -0.13 ppb m s$^{-1}$, respectively. It indicates the transportation contribution was low.

HCHO strongly correlated with proxy radicals. Comparing to the HCHO vertical distribution at the same time, we found $O_3$ and HCHO have a similar vertical distribution and the high HCHO concentration (> 8 ppb) occurred prior to the increase of $O_3$ concentration. Previous study shows that the concentration of isoprene in southern part of Shanghai ranged from 1 to 6 ppb during summertime (Geng et al., 2011). The high HCHO are mainly contributed by the oxidation of biogenic emission from plants. Therefore, the $O_3$ measured is most likely coming from local formation under VOC-sensitive regime.

(5) Line 308 -309, "High HCHO levels during the time are mainly contributed by the oxidation of biogenic emissions of VOCs from plants, i.e., isoprene." Do the authors have evidences for such statement? At least a reference is needed for this statement.

R: Geng et al. (2011) reported that isoprene emissions in the southern part of Shanghai are higher than the northern part during summer. The emission rates and ambient concentrations of isoprene are expected to be high especially from 9:00 to 15:00 local time. As the campaign was performed during summertime, the meteorological conditions are favorable for the production of isoprene from plants (Guenther et al., 2000). We have cited the related reference and improved the statement in the manuscript (Line 317-319).

(6) Line 317 – 318, "Major sources of atmospheric aerosols in FengXian area could be the sea salt aerosols because the measurement site is close to the coast." This is probably not true. In China, even for the area close to the coast, the aerosols were normally dominated by sulfate, nitrate, ammonia and organics. I suggest the authors to look up the available published results of aerosol chemical composition in Shanghai and surrounding areas.

R: We have reviewed some previous studies, in general, sulfate, nitrate and ammonium together contributed to more than half of the total $PM_{2.5}$ through the year, while the fraction of sea salt particles were still a small part even it increased during clean days at Shanghai (Pathak et al., 2009; Tao et al., 2011; Han et al., 2015). This information is included in the manuscript now (Line 328-330).

(7) The judgement of the secondary aerosol formation from a correlation of $NO_2$ and aerosol extinction coefficient seems to be quite arbitrary. And R = 0.63 cannot be well recognized as a significant correlation. More discussions are required to support this deduction. And the information by Figure 11 is very limited, I suggest to move this figure into supporting materials.

R: The investigation of secondary aerosol formation is mainly based on analyzing the correlation between precursor gases and aerosol concentrations combining multiple observational, experimental and modeling methods with consideration of other influencing factors like meteorological conditions. In this study, we combined several different measured techniques focusing on the air pollutants vertical profiles and possible reason for ozone formation. However, the measurement methods used in this study are mostly based on remote sensing technique. They can provide height resolved information of aerosols and trace gases, while detailed analysis of the chemical composition of aerosol is impossible. The concentration of $SO_2$ (R 0.6~0.9 in winter) and $NO_2$ (R=0.63) were correlated with aerosol extinction in order to qualitative analyze the relationship between precursor gases and the contribution of secondary formation of particles in previous studies (Wang et al. 2014; Wang et al., 2017). Therefore, we can only focus on the temporal and spatial correlation between aerosols and trace gases in order to estimate the aerosol formation pathway. A more detailed description and explanation is now included in the manuscript Line 339 to 346. Nevertheless, we fully agree that more information are needed for the investigation of the secondary aerosol formation and make a better layout of the observation plan in the future study.

In addition, we accepted the reviewer comment and moved the Figure 11 to supporting materials part.

Technical comments:
Line 168, change 'Dependency' to be 'Dependence'
Line 227, USTC shall be explained when appeared for the first time.
Line 580, Figure caption of Figure 2, (a) . . . as well as the correlation coefficients shown in the brackets. (b) . . . shows the mean differences and standard deviations in

the brackets . . .

Line 626, Spatial distribution of USTC OMI tropospheric NO2 . . .

R: We have followed these suggestions and made the corrections accordingly.

**List of changes in the manuscript**

-line 75: Lidar → lidar

-line 104-106: Add "Though lower elevation angles are more sensitive to the near ground aerosols and trace gases, the lowest elevation that we could reach is 3° due to the complex topography surrounding the measurement site."

-line 107: calculated depending on → adjusted according to

-line 109: using → by

-line 110: delete "of"

-line 111: stratosphere → stratospheric absorbers

-line 136: Add ",which can cover at least one full scan sequence and include all the measured DSCDs during this period."

-line 163-166: Add "The $O_3$ concentrations were measured by a UV photometric analyzer (Thermo 49i, Thermo Fisher, USA). $NO_2$ concentrations were obtained from a nitrogen oxides analyzer (Thermo 42i, Thermo Fisher, USA). Moreover, particulate matters ($PM_{2.5}$ and $PM_{10}$) concentrations were measured with the tapered element oscillating 165 microbalance (TEOM) online particle monitors (Thermo TEOM 1405, Thermo Fisher, USA)."

-line 174: Dependency → Dependence

-line 196: Add "of May 17, 2016"

-line 233: USTC → USTC (University of Science and Technology of China)

-line 247: The discrepancy → These discrepancies

-line 252: The → the

-line 278: Add "$O_3$ concentration between"

-line 279: 0.09 → 0.3

-line 291: 8:00 → 08:00

-line 299: Add ", resulting the lower $O_3$ vertical flux of -1.2, -1.2, -0.08 and -0.13 ppb m s$^{-1}$ at these altitudes."

-line 314: Add "both horizontal and vertical"

-line 317-318: Add "The isoprene emission in southern part was also reported to be higher than the northern part of Shanghai during summertime (Geng et al., 2011)."

-line 319: Add "with the favorable meteorological conditions,"

-line 327-330: "Major sources of atmospheric aerosols in FengXian area could be the sea salt aerosols because the measurement site is close to the coast." → "In Shanghai

area, sulfate, nitrate and ammonium together contributed to more than half of the total PM$_{2.5}$ through the year, while the fraction of sea salt particles increased during clean days (Pathak et al., 2009; Tao et al., 2011; Han et al., 2015)."

-line 339-346: "In order to qualitative the relationship between precursor gases and particles at different altitudes, correlation between NO$_2$ concentrations and aerosol extinctions from 100 m to 1000 m above ground level on 9 May is shown in Fig. S6. Moderate correlation is observed between ambient NO$_2$ and aerosol (R=0.63) indicating the feasible contribution of NO$_2$ to the secondary aerosol formation from the ground level to higher altitudes. Under high atmospheric ozone conditions, ambient NO$_2$ is oxidized rapidly to form nitrate aerosols in the atmosphere of Shanghai (Du et al., 2011). Nevertheless, more information like in-situ chemical composition and atmospheric conditions are needed for the investigation of the detailed 345 secondary aerosol formation pathway."

-line 357-358: Add "The improvement of USTC OMI NO$_2$ products is mainly related to the usage of localized NO$_2$ and aerosol vertical profiles for the AMF calculation."

-line 420: Add "Geng, F., Tie, X., Guenther, A., Li, G., Cao, J., and Harley, P.: Effect of isoprene emissions from major forests on ozone formation in the city of Shanghai, China, Atmos. Chem. Phys., 11, 10449-10459, 2011."

-line 429: Add "Han T., Qiao L., Zhou M., Qu Y., Du J., Liu X., Luo S., Chen C., Wang H., Zhang F., Yu Q. and Wu Q.: Chemical and optical properties of aerosols and their interrelationship in winter in the megacity Shanghai of China, J. Environ. Sci (China), 27, 59-69, 2015."

-line 485: Add "Pathak, R. K., Wu, W. S., and Wang, T.: Summertime PM$_{2.5}$ ionic species in four major cities of China: nitrate formation in an ammonia-deficient atmosphere, Atmos. Chem. Phys., 9, 1711-1722, 2009."

-line 509: Add "Tao S., Wang X., Chen H., Yang X., Li M., Li L. and Zhou Z.: Single particle analysis of ambient aerosols in Shanghai during the World Exposition, 2010: two case studies, Frontiers of Environmental Sciences& Engineering in China, 4, 391-401, 2011."

-Table 2: Add "on May 17, 2016" in the title

-Figure 2: Add "shown in the brackets" and "in the brackets" in the caption.

-Figure 4: Add "USTC" in the caption

-Figure 6: Change the figure time format axis

-Figure 11: move it to Supplement as Fig. S6

[revised manuscript text omitted]